# Effect of Leflunomide–Metal Complexes on ROS, TNF, and Brain Indolamines in Comparison with Anti-Depressants as Adjunct Therapy in Rheumatoid Arthritic Model

**DOI:** 10.3390/biomedicines11082214

**Published:** 2023-08-07

**Authors:** Almas Naeem, Noor Jahan, Moona Mehboob Khan, Ghulam Abbas, Faheema Siddiqui, Muhammad Usaid Khalid, Waqas Ahmed Farooqui

**Affiliations:** 1Department of Pharmacology, Dow College of Pharmacy, Dow University of Health Sciences, Karachi 74200, Pakistan; almasnaeem6@gmail.com (A.N.); faheema.siddiqui@duhs.edu.pk (F.S.); rph.m.usaid.khalid@gmail.com (M.U.K.); 2Department of Pharmaceutical Chemistry, Dow College of Pharmacy, Dow University of Health Sciences, Karachi 74200, Pakistan; moona.mehboob@duhs.edu.pk; 3Department of Pharmacology, Ziauddin University, Karachi 75000, Pakistan; ghulam.abbas@hotmail.com; 4School of Public Health, Dow University of Health Sciences, Karachi 74200, Pakistan; waqas.ahmed@duhs.edu.pk

**Keywords:** leflunomide, metal complexes, rheumatoid arthritis, adjuvant-induced arthritis, fluoxetine

## Abstract

Leflunomide is an isoxazole immunomodulating drug used to treat rheumatoid arthritis (RA). It is adopted as a metal-containing molecule to proceed with saturated salts of essential and detected metals; it amends the pharmacokinetic and pharmacodynamics activity of leflunomide to provide [M(Lef)_4_]X_2_-type complexes. Earlier it has been reported that after forming complexes with metals, leflunomide anti-arthritic activity was significantly altered in an acute arthritic model. In the present study, we evaluated the possible modification in anti-arthritic activities of leflunomide–metal complexes (Mg^+2^, Ca^+2^, Fe^+2^, Zn^+2^) with and without an anti-depressant drug, i.e., fluoxetine (10 mg/kg) in a chronic AIA model. Rats (n = 5) were administered with 0.1 mL of CFA into the right hind paw while treated groups received leflunomide and its metal complexes orally (3.2 mg/kg) for 24 days. On the final day of experiment, rats were sacrificed; a specific rat immunoassay ELISA kit was used to assess TNF-α in serum samples and read at 450 nm; a tissue sample of a paw was homogenized in a phosphate buffer using DCFH-DA dye for binding to assess ROS. A rat’s brain sample was homogenized and evaluated for tryptophan, serotonin (5-HT), and HIAA by RP-HPLC with EC detector. The overall TNF production was altered in treated rats. In addition, a decreased ROS was observed in all categories, except lef+Mg^+2^ group. Moreover, depletion in the brain indolamine levels were found in treated groups; an upraised level of these indolamines was observed when fluoxetine was added. It is concluded that metals affect leflunomide activity on complexation and simultaneous administration of fluoxetine cope up with the depression in arthritic-induced rats.

## 1. Introduction

Rheumatoid arthritis (RA) is a chronic autoimmune disorder that influences 1% of the population worldwide. Epidemiological studies of RA indicated a consistent increase in mortality rate in the general population [1]. The prevalence rate of RA in general population of Pakistan is about 0.55% [2]. The frequency of RA in females is higher than males [3]. RA is due to hyperplasia of T and B lymphocytes. Moreover, levels of scavenger cells and cytokines, such as IL-1, IL-6, TNF, and ROS, are also altered, which plays a vital part in RA pathology [1,2,3,4,5,6]. Neurobiological disorders, such as depression, are very common in RA patients [3]. It is linked with prominent pro-inflammatory cytokines (IL-1, IL-6, TNF and ROS) and an increase in the catabolism of tryptophan through indoleamine-dioxygenase enzyme (IDO) activation that is released from interferon-gamma present in inflammatory cells. Tryptophan is the precursor of serotonin (the neurotransmitter involved in consciousness, mood elevation, and sleep); therefore, a reduction in tryptophan concentration decreased the synthesis of serotonin in the brain, which makes arthritic patients depressed [7,8,9,10,11,12]. To treat RA, leflunomide is a key DMARD, which is a pyrimidine inhibitor that inhibits the dihydroorotate dehydrogenase (DHODH) enzyme to prevent joint inflammation. Earlier it has been reported that leflunomide anti-arthritic activity was significantly altered in an acute arthritic model after forming complexes with metals [13]. Previously, in one of our studies, it has been reported that brain indolamine levels (tryptophan, serotonin, HIAA) decreased in adjuvant-induced arthritic (AIA) rats [4,5]. Later on, we also reported that not only serotonin but all brain indolamine levels (tryptophan, serotonin, HIAA) decreased in (AIA) rats [10], which were aggravated more when treated with leflunomide, which indicates that in RA one of the reasons for depression is decreased brain indolamine levels [14]. Our other study showed that administration of anti-depressant drugs, such as fluoxetine and imipramine, along with leflunomide in carrageenan-induced arthritic rats improves the depressive condition of RA [15]. Presently, we studied the anti-arthritic effect of leflunomide–metal complexes on RA and its associated depression in chronic an AIA model with and without an anti-depressant drug, i.e., fluoxetine, a selective serotonin reuptake inhibitor (SSRI), as depression therapy in RA patients is frequently ignored, negatively impacting patients’ quality of life. Therefore, in this study, we wanted to highlight that treating RA should be managed on both levels, i.e., to reduce the joint pain as well as its associated depression. Therefore, fluoxetine is selected, which is safe and efficacious in multiple rodent models of depression. This study was conducted on AIA rats. It is the most often used animal model in RA and predictor of multiple drugs’ clinical efficacy, having almost the same pathological conditions as humans who have RA [5].

## 2. Materials and Methods

### 2.1. Animals

Healthy female Wistar rats, 180–230 g, (9–10 weeks old) were obtained from the Animal House of Dow University of Health Sciences Karachi. The animals were housed in a room at 27 to 30 °C and a 12 h light/12 h dark cycle, while moisture was sustained at 40–70%. Free access to feed pellets and tap water was provided for laboratory animals. The study followed ethical guidelines of an international association [16]. The study was approved by the institutional ethical committee of Dow University of Health Sciences (Ref: IRB-1088/DUHS/Approval/2018/109).

### 2.2. Induction of Arthritis

An induction of arthritis with modification was carried out [17,18,19]. Arthritis was induced by adjuvant for a chronic model of arthritis. In each category, rats (n = 5) were used. For this purpose, 0.1 mL of complete Freund’s adjuvant (CFA) (1 mg/mL of heat-killed mycobacterium tuberculosis) (Invitrogen Co, Carlsbad, CA, USA), with 0.1 mL of paraffin oil was injected via sub-planter route into the right hind paw. This day was considered as day zero. All arthritic groups received their respective treatment on the same day except the AIA control group. For the treatment, API (active pharmaceutical ingredient) of leflunomide and fluoxetine was purchased from Sigma company (SigmaAldrich, Germany). Leflunomide–metal complexes were synthesized in the research lab of pharmaceutical chemistry, Faculty of Pharmaceutical Sciences, Dow College of Pharmacy. Synthesis was carried out as described earlier [13]. Briefly, the job’s method of continuous variation was used to estimate the stoichiometric ratio between leflunomide and metal salts. Leflunomide:metal ratio of 4:1 produced the strongest interaction. Then, separately, the solution of hydrated salts of each metal (Mg++, Ca++, Fe++, and Zn++) was made in methanol and combined with the leflunomide–methanolic solution. These solutions were refluxed at 80 °C for a couple of hours. TLC was used to monitor the complex’s synthesis throughout the experiment to pinpoint its completion. The solution was then allowed to crystallize. Crystals were characterized using spectroscopic methods, such as IR, NMR, and CHN elemental analyses. Metal complexes were utilized in this work once their production was verified [13].

### 2.3. Animal Groups

Animals were grouped into twelve groups, i.e., normal saline-treated healthy rats (normal), non-treated arthritic rats (AIA negative control), leflunomide-treated arthritic rats (AIA-lef positive control), lef–magnesium complex-treated arthritic rats (AIA-lef+Mg⁺^2^), lef–calcium complex-treated arthritic rats (AIA-lef+Ca⁺^2^), lef–ferrous complex-treated arthritic rats (AIA-lef+Fe⁺^2^), lef–zinc complex-treated arthritic rats (AIA-lef+Zn⁺^2^), leflunomide- and fluoxetine-treated arthritic (AIA-lef+flox) group (positive control), leflunomide–magnesium complex- and fluoxetine-treated arthritic rats (AIA-lef+Mg⁺^2^+flox), leflunomide–calcium complex- and fluoxetine-treated arthritic rats (AIA-lef+Ca⁺^2^+flox), leflunomide–ferrous complex- and fluoxetine-treated arthritic rats (AIA-lef+Fe⁺^2^+flox) and leflunomide–zinc complex- and fluoxetine-treated arthritic rats (AIA-lef+Zn⁺^2^+flox). The number of rats in each group was 5.

### 2.4. Anti-Inflammatory Activity

Treatment of the respective groups with leflunomide and leflunomide–metal complexes were carried out at a 3.2 mg/kg dose PO for 24 days. Protocol commenced on the day arthritis was induced and considered as day zero; then, it continued for 24 days (the day when full arthritis was developed in rats of the AIA control group, as indicated by observing macrocsopic analysis of their joints). Fluoxetine was used at 10 mg/kg orally to study the dose–response relationship. An assessment of anti-inflammatory activity was performed by macroscopic analysis. Each paw score from zero to four supported an extent of erythema, swelling, and distortion of the joints: zero = normal; one = slight erythema; two = erythema and swelling of 2 toes or fingers; three = severe erythema and swelling of the ankle or wrist; and four = complete erythema and swelling of toes or fingers, joint deformity, and lack of flexibility [20].

Paw volume was also measured by plethysmometer, an instrument used to assess the inflammatory response in rodents. This instrument showed clear, three-dimentional picture of the tibiotarsal joint. Animals’ body weight was also observed on alternate days during the whole experiment [17].

### 2.5. Analysis of TNF-α and ROS

At day 24, the final day of experiment when complete AIA developed in control group, all female Wistar rats were sacrificed, and serum samples were used for the evaluation of the tumor necrosis factor (TNF) and tissue sample of the paw that was used for the evaluation of the reactive oxygen species (ROS). A specific rat immunoassay ELISA kit was used to assess TNF-α according to the producer’s recommendations. Levels of TNF-α were measured by using an Invitrogen ^TM^ Rat TNF-α ELISA kit (Invitrogen Co., Carlsbad, CA, USA). The extent of TNF-α in serum samples was observed from standard curves at an absorbance of 450 nm. A tissue sample of the paw was integrated in a phosphate buffer (pH 7.27) and then centrifuged (14,000× *g*) for 5 min. The supernatant (50 µL) was incubated in the dark for a period of 30 min at 37 °C, with 2′,7′-dichlorodihydrofluorescein diacetate (DCFH-DA) (50 µL, 20 mM) in 96 black wells plate. The DCFH-DA dye was attached with ROS and turned into a highly fluorescent 2′,7′dichlorodihydro-fluorescein (DCF) that was interpreted at a wavelength of 485 nm of excitation and 538 nm of an emission-employing fluorescence microplate reader [21,22].

### 2.6. Brain Dissection Technique

Brain indolamines levels tryptophan (TRP), serotonin (5-HT), and hydroxy indole acetic acid (HIAA) were estimated by rats’ brain samples. When rats were sacrificed, their brains were immediately excised from cranial cavity by removing them with a durameter and stored at −80 °C on the final day of the experiment [14].

### 2.7. Extraction of Indolamines from the Brain

An electric homogenizer was used for extraction of indolamines from the rat’s frozen brain sample as previously described and an assessment of tryptophan, serotonin (5-HT), and HIAA was performed by RP-HPLC coupled with an EC detector because of very low concentration of indolamines in the brain sample [14].

### 2.8. Estimation of Brain Indolamines

A reverse-phase HPLC method coupled with an electrochemical detector was operated to obtain an estimation of the concentration of these indolamines from the brain samples. Their levels were assessed as formerly reported [23] to investigate the impact of therapy on RA-related depression. Quantification of indolamines was attained by the RP-HPLC (Prominence 20A, Shimadzu, Japan) system comprised of a pump (LC-20A), an autosampler (SIL-20A), and an electrochemical detector (L-ECD-6A) attached with a carbon electrode and modulated at a potential of +0.75 V. The communication bus module (LC-CBM-20A) attached the hardware with LC 6.0 software. The HPLC buffer (sodium dihydrogen citrate anhydrous (0.1 M), 1-octane sulfonic acid (2.5 mM), and ethylenediaminetetraacetic acid (EDTA) disodium salt (1 mM) carried out 12% acetonitrile, pH 3.6) was filtered (0.22 mm) and removed with helium. The reversed-phase Nucleosil column (C18, 250/4.6 mm, 5 mm) heralded by a guard column (C18, 8/4 mm, 5 mm, Macherey Nagel Inc., Allentown PA, USA) was used [24].

### 2.9. Statistical Analysis

Results were analyzed statistically by one-way ANOVA accompanied by Tukey’s post hoc test, using SPSS 21 software at a significance level of *p* < 0.05 and *p* < 0.005. n = 5 in each group. Results are expressed in mean ± SD and * = significant at *p* < 0.05; ** = significant at *p* < 0.005 in comparison to the control group.

## 3. Results

Leflunomide and its metal complexes were studied in female rats with oral administration at the regular dose of 3.2 mg/kg for 24 days. The parent and its metal complexes were also compared with an addition of the antidepressant drug fluoxetine, as it is a known fact that anti-arthritic medications induce depression. An anti-arthritic effect was observed by observing a paw edema by macroscopic analysis. Results were analyzed at a significance level of *p* < 0.05 when compared with the control group. The studies showed that some of metal complexes as a zinc–leflunomide complex gave a better response than the parent compound, i.e., leflunomide (Figure 1 and Figure 2). It was also observed that the addition of fluoxetine with leflunomide or its metal complexes did not decrease the anti-arthritic effect, which showed no interaction of leflunomide and its metal complexes with fluoxetine. An anti-arthritic effect of an iron–leflunomide complex is somewhat similar to the parent, i.e., leflunomide. Two other metal complexes, i.e., magnesium and calcium complexes, also gave an anti-arthritic effect, but it was somewhat less than the parent compound (Figure 1 and Figure 2). The order of anti-inflammatory activity was AIA-lef > AIA-lef+Zn^+2^ > AIA-lef+Fe^+2^ >AIA-lef+Ca^+2^ > AIA-lef+Mg^+2^ > AIA control. The same order was observed in combination with fluoxetine.

Weight variation of the rats was observed throughout the experiment. No prominent weight reduction was observed in all the twelve groups (Figure 3). A slight increase in weight was observed in the rats receiving the iron–leflunomide complex.

### 3.1. TNF-α alongside and without Fluoxetine

The level of TNF-α in groups not treated with fluoxetine showed a significant (*p* < 0.0001, F = 899.668) difference when compared with the normal group (Figure 4). The order of anti-inflammatory activity was AIA-lef > AIA-lef-Ca^+2^ > AIA-lef-Zn^+2^ > AIA-lef-Fe^+2^ > AIA-lef-Mg^+2^ > AIA control. The results of the fluoxetine groups also showed a significant (*p* < 0.0001, F = 948.364) difference among all the groups of animals judged against the normal group (Figure 4). The order of anti-inflammatory activity was AIA-lef+flox > AIA-lef-Fe^+2^+flox > AIA-lef-Ca^+2^+flox > AIA-lef-Mg^+2^+flox > AIA-lef-Zn^+2^+flox > AIA control.

### 3.2. ROS with and without Fluoxetine

In groups not receiving fluoxetine, the levels of ROS were altered significantly (*p* < 0.0001, F = 539.551) when compared with the normal group (Figure 5). Data indicated that the best antioxidant activity to control the ROS was observed in lef-Zn group when compared with all other groups. The addition of fluoxetine with these metal complexes also indicated an alteration in activity to control ROS in all groups. The order of anti-inflammatory activity was AIA-lef-Zn^+2^ > AIA-lef > AIA-lef-Fe^+2^ > AIA-lef-Ca^+2^ > AIA-lef-Mg^+2^ > AIA control. Results with the fluoxetine group also showed a significant difference (*p* < 0.0001, F = 961.652) among all the groups of animals in contrast to the normal group. Results revealed no difference (*p* > 0.05) between the leflunomide–metal complex of Zn^+2^+flox and the negative control AIA group, which showed increased production of ROS in this group. The order of anti-inflammatory activity was AIA-lef+flox > AIA-lef-Ca^+2^+flox > AIA-lef-Mg^+2^+flox > AIA-lef-Fe^+2^+flox > AIA-lef-Zn^+2^+flox > AIA control (Figure 5).

### 3.3. Indolamines Determination

#### 3.3.1. Drugs’ Effect on Brain Indolamines (Tryptophan, Serotonin, and Hydroxyindole Acetic Acid) without the Fluoxetine Group

In this study, levels of brain indolamines were reduced significantly (*p* < 0.005) in all arthritic groups relative to normal rats, which elucidated depression in every arthritic groups as indicated in Figure 6, Figure 7 and Figure 8. This investigation also designated that the therapy of RA in AIA rats showed a noticeably and exceedingly significant (*p* < 0.005) effect on brain tryptophan (F = 1,056,392.713, *p* < 0.005), serotonin (F = 16,990.785, *p* < 0.005), and HIAA (F = 7579.472, *p* < 0.005) levels. The post hoc Tukey’s test for inter group mean differences depicted that this depression amplified significantly (*p* < 0.005) when these arthritic rats were served with the leflunomide positive control group and leflunomide–metal complexes, i.e., (Mg^+2^, Ca^+2^, Fe^+2^, Zn^+2^). Furthermore, severe depletion in brain indolamine concentration was noticed in the treatment groups. The order of reduction in tryptophan levels was AIA-lef-Ca^+2^ > AIA-lef-Mg^+2^ > AIA-lef-Fe^+2^ > AIA-lef-Zn^+2^ > AIA control > AIA-lef. The order of reduction in serotonin levels was AIA-lef-Ca^+2^ > AIA-lef-Mg^+2^ > AIA-lef-Fe^+2^ > AIA-lef-Zn^+2^ > AIA control > AIA-lef. The order of reduction in HIAA levels was AIA-lef-Ca^+2^ > AIA-lef-Mg^+2^ > AIA-lef-Fe^+2^ > AIA-lef-Zn^+2^ > AIA control > AIA-lef.

#### 3.3.2. Drugs’ Effect on Brain Indolamines (Tryptophan, Serotonin, and Hydroxyindole Acetic Acid) alongside the Fluoxetine Group

In this study, brain indolamine levels manifested a significant (*p* < 0.005) difference in all the groups of leflunomide with fluoxetine–metal complexes of Mg^+2^, Ca^+2^, Fe^+2^, and Zn^+2^ relative to the normal control rats as indicated in Figure 9, Figure 10 and Figure 11. This investigation also suggested that the treatment of RA in AIA rats created a significant (*p* < 0.005) effect on brain tryptophan (F = 533,073.218, *p* < 0.005), serotonin (F = 3535.806, *p* < 0.005), and HIAA (F = 1900.510, *p* < 0.005) levels. The post hoc Tukey’s test for inter group mean differences conferred a significant (*p* < 0.005) difference in all groups relative to the normal control group, except the AIA-lef-Fe^+2^+flox group; this group showed an insignificant difference (*p* > 0.005) in HIAA levels and produced an adequate anti-depressant effect. The order of increase in tryptophan levels and anti-depressant activity was AIA-lef-Fe^+2^+flox > AIA-lef+flox > AIA-lef-Ca^+2^+flox > AIA-lef-Mg^+2^+flox > AIA-lef-Zn^+2^+flox > AIA control. The order of rise in serotonin levels and anti-depressant activity was AIA-lef-Fe^+2^+flox > AIA-lef+flox > AIA-lef-Ca^+2^+flox > AIA-lef-Zn^+2^+flox > AIA-lef-Mg^+2^+flox > AIA control. The order of increase in HIAA levels and anti-depressant activity was AIA-lef-Fe^+2^+flox > AIA-lef+flox > AIA-lef-Ca^+2^+flox > AIA-lef-Zn^+2^+flox > AIA-lef-Mg^+2^+flox > AIA control.

## 4. Discussion

Previous studies showed some dose-response relationship in an acute model, specifying that metals influence leflunomide activity on complexation [13]. In the present work, leflunomide–metal complexes were considered for their efficacy on RA treatment using a chronic arthritic model. Altered levels of TNF-α have been found in the serum of RA patients, which was held accountable for inflammation and joint destruction in arthritic patients. [25]. Some studies showed levels of TNF-α and IL-6 both became elevated in the serum of RA patients [26,27]. In the current study, serum levels of TNF-α were determined to evaluate the severity of rheumatoid arthritis. The overall TNF production was elevated in all leflunomide–metal complexes without fluoxetine-treated rats, and dysregulation of TNF production was observed in all leflunomide–metal complexes with fluoxetine-treated rats. InRA, generation of ROScauses tissue damage [28]. Overproduction of these pro-inflammatory cytokines and chemokines release inflammatory mediators, and activated macrophages produce ROS that leads to tissue damage [29,30]. The overall decreased production of ROS in all leflunomide–metal complexes without fluoxetine showed controlled tissue damage in RA, except the leflunomide–metal complex of the Mg^+2^ group, which showed uncontrolled tissue damage and joint destruction in RA. Fluoxetine ensures opposition to the adverse effects of various sorts of stressors [31,32]. Moreover, it lessens the impact of stress on the immune system [33,34] and provides protection in the case of oxidative damage [35,36,37]. Along with fluoxetine, overall ROS production was decreased and showed controlled tissue damage in RA, except the leflunomide–metal complex of the Zn^+2^ +flox group and negative control AIA group.

Depletion of brain indolamines is a root cause of depression. Previous studies showed depression and anxiety produced by the depletion of brain serotonin levels [23,38]. Leflunomide–metal complex of Mg^+2^, Ca^+2^, Fe^+2^, and Zn^+2^ showed a decrease in tryptophan, serotonin, and HIAA levels, and because of degradation of brain indolamines, it led to depression in all animals. Hence, the decrease in availability of tryptophan in brain is due to low brain serotonin levels that play a part in decreased serotonin synthesis at this region [39]. Our results specify that decreased serotonin, tryptophan, and HIAA levels in the brain may be one of the significant causes that make RA patients depressed [23]. Groups that received fluoxetine in combination with leflunomide–metal complex of Mg^+2^, Ca^+2^, Fe^+2^, and Zn^+2^ showed an increase in tryptophan, serotonin, and HIAA levels, and less degradation of brain indolamines was observed, which showed fluoxetine produced its anti-depressant effect. Fe^+2^ +flox showed an overall increase in tryptophan, serotonin, and HIAA levels, which showed less depression was observed in this group.

## 5. Conclusions

The effect of leflunomide and its metal complexes at a dose of (3.2 mg/kg/day PO) produced a potent therapeutic action that showed its ability to significantly reduce inflammation in adjuvant-induced arthritic rats. It has been concluded that leflunomide and its metal complexes are optimistic anti-arthritic agents in the treatment of inflammatory disorders. Earlier studies showed rheumatoid arthritis cases not only experience joint pain but also depression, which are exaggerated with leflunomide [14]. Leflunomide–metal complex with Zn produced more promising results in comparison to others. The available study also discloses that simultaneous administration of fluoxetine with leflunomide and its metal complexes cope up with the depression without affecting its anti-arthritic efficacy.

## Figures and Tables

**Figure 1 biomedicines-11-02214-f001:**
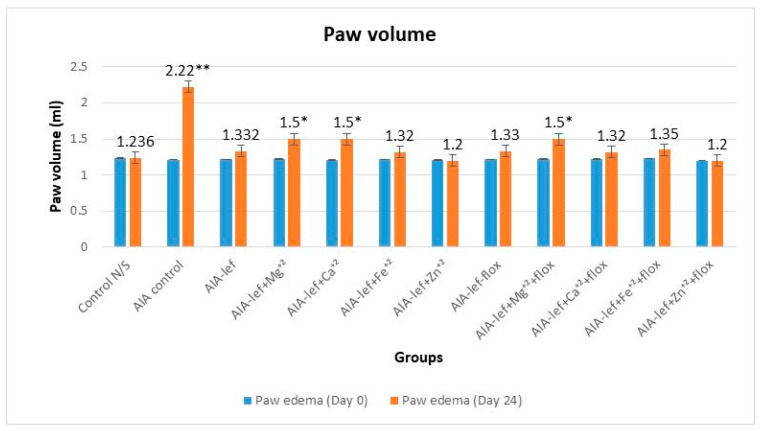
Effect of test compounds on paw edema of rats at day 24. * = significant at *p* < 0.05; ** = significant at *p* < 0.005 in comparison to the control group.

**Figure 2 biomedicines-11-02214-f002:**
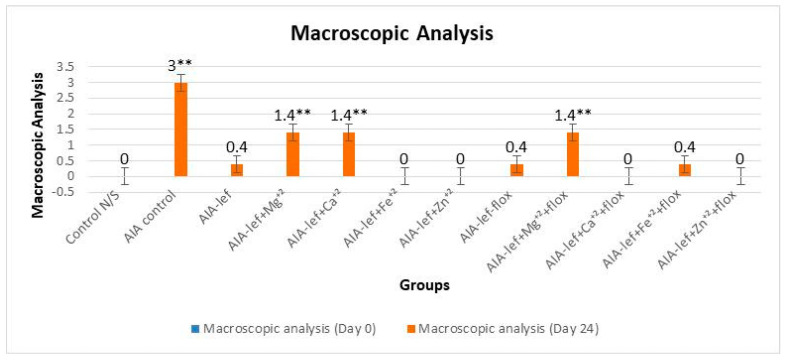
Macroscopic analysis of a rat’s paw at day 24 after administration of test compounds. ** = significant at *p* < 0.005 in comparison to the control group.

**Figure 3 biomedicines-11-02214-f003:**
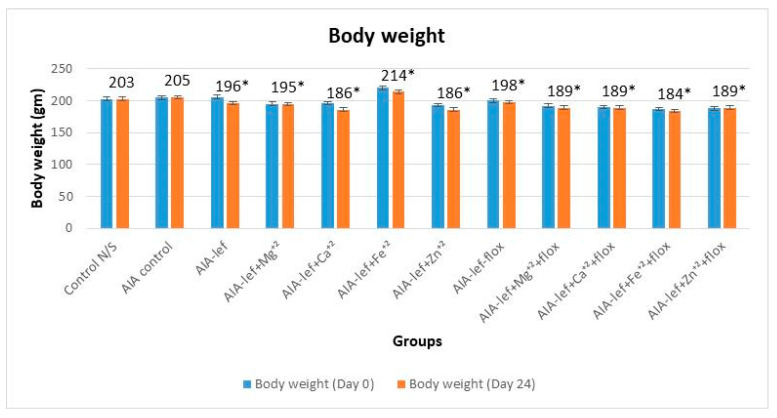
Body weight of rats at day 24 of treatment. * = significant at *p* < 0.05 in comparison to the control group.

**Figure 4 biomedicines-11-02214-f004:**
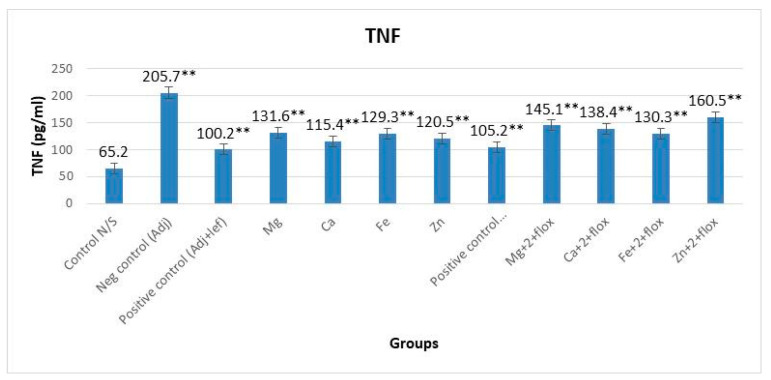
Effect of test compounds on level of TNF-α. ** = significant at *p* < 0.005 in comparison to the control group.

**Figure 5 biomedicines-11-02214-f005:**
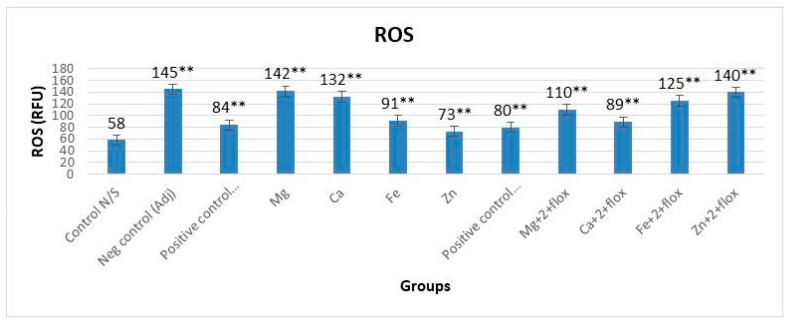
Effect of test compounds on level of ROS. ** = significant at *p* < 0.005 in comparison to the control group.

**Figure 6 biomedicines-11-02214-f006:**
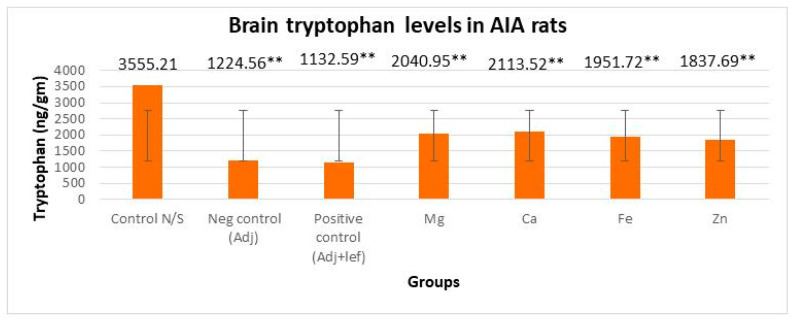
Effect of test compounds on level of brain indolamines levels (tryptophan). ** = significant at *p* < 0.005 in comparison to the control group.

**Figure 7 biomedicines-11-02214-f007:**
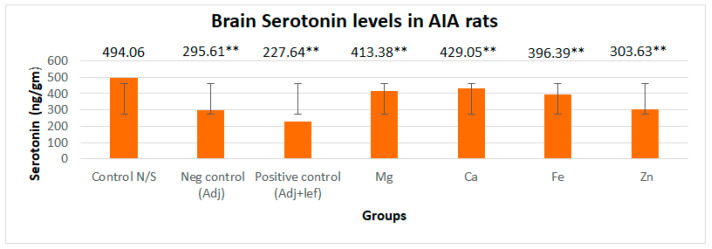
Effect of test compounds on brain indolamines levels (serotonin). ** = significant at *p* < 0.005 in comparison to the control group.

**Figure 8 biomedicines-11-02214-f008:**
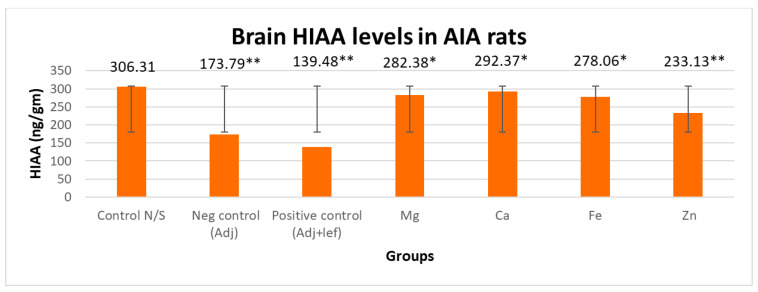
Effect of test compounds on brain indolamines levels (HIAA). * = significant at *p* < 0.05; ** = significant at *p* < 0.005 in comparison to the control group.

**Figure 9 biomedicines-11-02214-f009:**
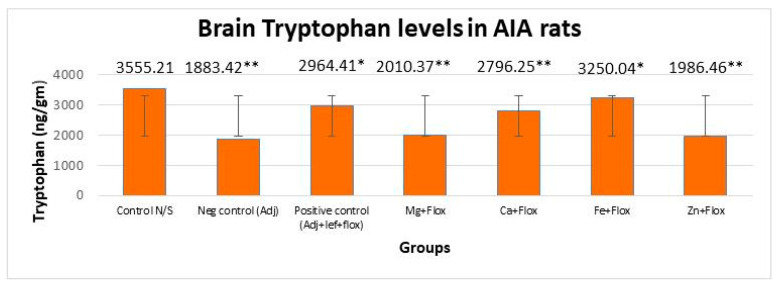
Effect of test compounds with fluoxetine on brain indolamines levels (tryptophan). * = significant at *p* < 0.05; ** = significant at *p* < 0.005 in comparison to the control group.

**Figure 10 biomedicines-11-02214-f010:**
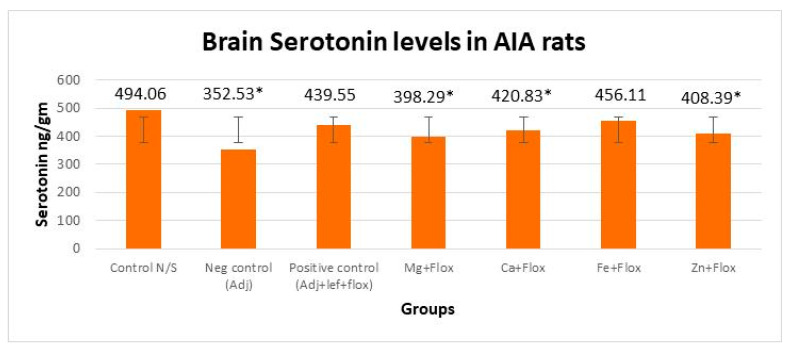
Effect of test compounds with fluoxetine on brain indolamines levels (serotonin). * = significant at *p* < 0.05 in comparison to the control group.

**Figure 11 biomedicines-11-02214-f011:**
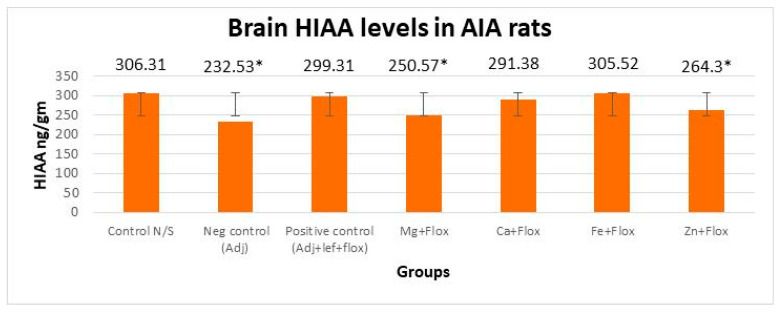
Effect of test compounds with fluoxetine on brain indolamines levels (HIAA). * = significant at *p* < 0.05 in comparison to the control group.

## Data Availability

Data available in a publicly accessible repository.

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
