# Peer review of "Effect of Leflunomide–Metal Complexes on ROS, TNF, and Brain Indolamines in Comparison with Anti-Depressants as Adjunct Therapy in Rheumatoid Arthritic Model"

_biomedicines, 2023, doi:10.3390/biomedicines11082214_

Round 1

Reviewer 1 Report

In the current article the authors presented the study of the antiarthritic effect of leflunomide metal complexes on RA and its associated depression in chronic AIA model with and without an anti-depressant drug (fluoxetine). They concluded that leflunomide and its metal complexes can be used as an antiarthritic agent in the treatment of inflammatory disorders.

Comments:

Introduction:

-Line 41- Rheumatoid Arthritis = RA. Complete please

-Lines 50-51: Give please some details concerning the statement: Previously one of our study has reported that brain indolamine levels (tryptophan, serotonin, HIAA) decreased in AIA rats.

Materials and methods:

- the method by which you obtained the leflunomide metal complexes is not presented. Or, where from you have them? Please add. -add also de origin of fluoxetine

- rewrite please the paragraph 2.2 Induction of arthritis. It’s not clear

-line 76 – add please how many mice you used/group

Results:

You wrote that you determined tryptophan, serotonin (5-HT) and HIAA using RP-HPLC coupled with EC detector. Please  present the obtained results as Suppl Mat. You presented the results in Figures, but in my opinion is not enough.

Moderate editing of English language required

Reviewer 2 Report

The present article presents the results of an experiment on laboratory animals that follows the evaluation of effect of leflunomide metal complexes on ROS, TNF and brain indolamines in comparison with anti-depressants adjunct therapy. This is a very interesting study idea.

Here are some remarks:

-line 19 please redefine leflunomide from immunoreactive drug to immunomodulatory drug

-line 19 – rheumatoid arthritis should be abbreviated as RA only after this is mentioned

-line 41 – RA is an “immedicable” autoimmune disorder – sounds quite strange and inaccurate, please modify according to actual definitions, also please provide references for mentioned epidemiology and immunopathology data

-line 44 – “RA patients also sufers with its associated depression” – please correct grammar errors and provide more information and reference

-line 48 – DHODH enzyme is dihydroorotate dehydrogenase

-lines 56 & 58 please do not use abbreviations (AIA) before it’s explanation

Material and method - for the assessment of joint swelling a single evaluator or two/more evaluators were used? This is important because the macroscopic evaluation is quite subjective.

How was the 24-day duration of the protocol chosen?

-Lines 283-285 please specify the references that link leflunomide with depression

Round 2

Reviewer 1 Report

Materials and methods:

- the method by which you obtained the leflunomide metal complexes is not enough presented.  

Results:

-As Suppl Material in my opinion you must add the HPLC spectra

Minor editing of English language is required

Author Response

Response to Reviewer 1 Round 2 Comments

Point 1: Materials and methods:

- the method by which you obtained the leflunomide metal complexes is not enough presented.  

Response 1: It is now further explained. Reference 13 is our coauthor paper, kindly consider the reference also.

Point 2: Results:

-As Suppl Material in my opinion you must add the HPLC spectra

Response 2: kindly consider the appeal regarding this point.

Point 3:

Minor editing of English language is required

Response 3: Minor editing has been done.

Reviewer 2 Report

The authors revised the article according with suggestions. 

Author Response

Thank you for the Review and Recommendation.

Round 3

Reviewer 1 Report

The manuscript has been significantly improved. The authors addressed all the required issues and in my opinion, the manuscript is suitable for publication.

Author Response

Thank you for review and consideration.